



# Process Analysis of Elevated Concentrations of Organic Acids at Whiteface Mountain, New York

Christopher Lawrence [1], Mary Barth [2], John Orlando [2], Paul Casson [1], Richard Brandt [1], Dan Kelting [3], Elizabeth Yerger [3], and Sara Lance [1]

[1]Atmospheric Sciences Research Center (ASRC), University at Albany, SUNY ETEC building, 1220 Washington Ave, Albany NY 12226, USA
[2]Atmospheric Chemistry Observations and Modeling Laboratory (ACOM), National Center for Atmospheric Research, Boulder, CO 80301, USA
[3]Paul Smith's College Adirondack Watershed Institute (AWI), P. O. Box 265, Routes 86 and 30, Paul Smiths NY 12970, USA

**Correspondence:** Christopher E. Lawrence (celawrence@albany.edu)

**Abstract.** Organic acids represent an important class of compounds in the atmosphere but there are many uncertainties in understanding their formation; in particular, few investigations have been carried out as to their sources in the Northeast U.S. Associated with a heat wave and pollution event on 1-2 July, 2018, unusually high concentrations of formic (HCOOH), acetic ($CH_3COOH$), and oxalic (OxAc) acid in cloud water were measured at the summit of Whiteface Mountain (WFM) in upstate

New York. To investigate the gas phase production of organic acids for this pollution event, this work uses a combination of the regional transport model WRF-Chem which gives information on transport and environmental factors affecting air parcels reaching WFM, the Lagrangian chemical box model BOXMOX, which allows analysis analysi of chemistry with different chemical mechanisms. Two chemical mechanisms are used in BOXMOX: 1) the Model for Ozone and Related chemical Tracers (MOZART T1), and 2) the Master Chemical Mechanism version 3.3.1 (MCM). The WRF-Chem results show that air parcels

sampled during the pollution event at WFM originated in central Missouri, which has strong biogenic emissions of isoprene. Many air parcels were influenced by emissions of nitrogen oxides ($NO_x$) from the Chicago Metropolitan Area. Ozonolysis of isoprene and related oxidation products were the major sources of HCOOH in both mechanisms. $CH_3COOH$ was produced from acetyl peroxy radical ($CH_3CO_3$) reacting with the hydroperoxy ($HO_2$) radical, with MCM producing up to 40% more $CH_3COOH$ under conditions of high isoprene and low $NO_x$ compared to MOZART T1. Both mechanisms underpredicted

HCOOH and and $CH_3COOH$ by an order of magnitude compared to measurements at WFM. A simple gas+aqueous box model was used to determine if cloud water chemistry could have had an appreciable impact on organic acid formation. Aqueous chemistry exacerbated the discrepancies of HCOOH by leading to a net depletion within cloud water. There were large disagreements in the production of glyoxal (a key precursor of OxAc) between the two gas-phase mechanisms, with MOZART T1 showing stronger daytime production under high $NO_x$ conditions, while MCM showed strong nocturnal production via

ozonolysis chemistry. The gas + aqueous model exhibited strong production of OxAc within cloud droplets, with glyoxal serving as an important precursor. The substantial differences between chemical mechanisms and between observations and models indicates that further studies are required to better constrain gas and aqueous production of low molecular weight organic acids.





## 1 Introduction

Organic acids are an important class of compounds in the atmosphere that can represent an important fraction of organic aerosol, comprising up to 52 % of the water soluble organic carbon mass. (Sorooshian et al., 2007; Miyazaki et al., 2009; Kawamura and Bikkina, 2016; Kawamura et al., 2017). Organic acids can also contribute a large fraction of the acidity in cloud and rain water, particularly in remote and rural regions (Pye et al., 2020), and may contribute to new particle formation (Zhang et al., 2004, 2017; Kumar et al., 2019). Additionally, there is a growing evidence that organic acids are important in partitioning ammonia ($NH_3$) into ambient aerosol (Tao and Murphy, 2019; Li et al., 2021) and cloud water (Lawrence et al., 2023). Organic acids are ubiquitously found throughout the atmosphere, measured in locations including the Arctic (Mungall et al., 2018; Feltracco et al., 2021), urban environments, (Souza et al., 1999; Avery et al., 2001), biomass burning smoke plumes (Chaliyakunnel et al., 2016), and forested areas (Fulgham et al., 2019; Eger et al., 2020). Despite their ubiquity and their growing chemical importance in many regions around the world, they are often not routinely included in studies monitoring the chemical composition of cloud and rain water and are rarely investigated in detail within modeling studies. To address these shortcomings, this study investigates the key processes that led to unusually high concentrations of organic acids measured in Whiteface Mountain (WFM) cloud water on July 1st, 2018.

Formic (HCOOH) and acetic ($CH_3COOH$) acids are typically the most abundant monocarboxylic acids found in the atmosphere (Paulot et al., 2011; Link et al., 2020). Primary sources of HCOOH and $CH_3COOH$ include soil emissions, (Mielnik et al., 2018), biomass burning (Chaliyakunnel et al., 2016) and even certain species of ants (Graedel and Eisner, 1988; Legrand et al., 2012). HCOOH and $CH_3COOH$ are also produced from the atmospheric oxidation of VOCs (Figure 1). It is thought HCOOH and $CH_3COOH$ are largely biogenic in origin but also known to have important anthropogenic sources regionally including fossil fuel combustion and volatile chemical products. In particular, the oxidation of isoprene and its related oxidation products are considered the most important precursor VOCs. Even though these acids are commonly found in the atmosphere, they are typically underpredicted by current gas phase mechanisms, especially HCOOH (Millet et al., 2015; Yuan et al., 2015; Chen et al., 2021), with the underlying causes remaining unclear.

More recent work has revealed that cloud droplets may act as an important medium for the formation of organic acids. Volatile but highly water soluble gases like glyoxal can dissolve into cloud droplets, where they subsequently oxidize to form dicarboxylic organic acids such as oxalic acid (OxAc) (Figure 1) that remain within the particle phase after the cloud droplets evaporate (Blando and Turpin, 2000; Lim et al., 2005; Warneck, 2005; Ervens et al., 2003; Tilgner and Herrmann, 2010). This process is especially important for the formation of dicarboxylic acids like OxAc as they have no known secondary gas phase sources, while primary emissions cannot explain their atmospheric concentrations (Yao et al., 2004). Despite the prevalence of this chemistry, these processes are often ignored or are oversimplified in chemical transport models.

At the summit of WFM in upstate NY, there is an historic cloud water monitoring program that has been operating since 1994. This program was initially focused on investigating the formation of two acid deposition species, sulfate ($SO_4^{2-}$) and



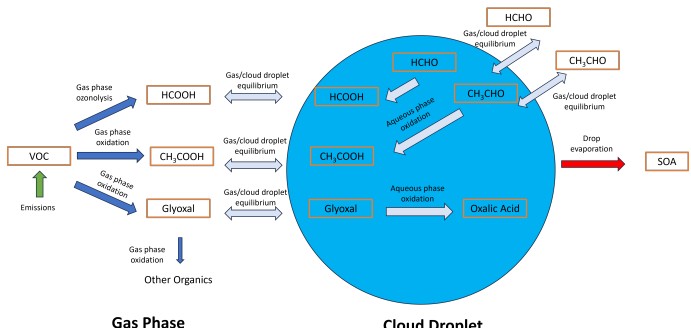

**Figure 1.** Summary of the major processes controlling organic acid production including emissions of VOCs, gas phase oxidation to form HCOOH and CH₃COOH and the important precursor glyoxal, gas/cloud equilibrium partitioning ,and the aqueous oxidation that either produces or removes organic acids. Important secondary organic aerosol chemistry is ignored to maintain simplicity of the schematic.

nitrate ($NO_3^-$), and was subsequently funded to monitor progress of the Clean Air Act Amendments of the 1990s. In more recent years, as the prevalence of acid deposition has decreased at WFM and throughout the United States, attention has shifted toward the organic fraction of cloud water (Schwab et al., 2016; Lawrence et al., 2023). Starting in 2018, organic acids were

added to the suite of regularly measured chemical species within cloud water which include HCOOH, CH₃COOH, and OxAc.

On July 1-2 2018, collected cloud samples exhibited unusually high concentrations of these organic acids with the underlying causes remaining unexplored. As the influence from $SO_4^{2-}$ and $NO_3^-$ in cloud water has decreased at WFM, at the same time that the influence from organic carbon has increased (Lawrence et al., 2023), the importance of organic acid contributions to the chemical system has grown, requiring a better characterization of the underlying chemistry. Chemical transport models can

be used to study the production of organic acids. However, it is challenging to investigate the major chemistry involved in their production upwind of a given location. Chemical box modeling can be used for a detailed look at the chemistry of organic acid production but the initial conditions and emissions of many chemical species, particularly VOCs, are limited both spatially and temporally. To overcome these limitations, a combination of chemical transport modeling and Lagrangian chemical box modeling can be used be to investigate organic acid production.

The current study used a combination of the chemical transport model Weather Research and Forecasting Model with Chemistry (WRF-Chem; Grell et al. (2005); Fast et al. (2006)) and the gas phase chemical box model BOXMOX (Knote et al., 2015) to evaluate the chemistry affecting the high concentrations of organic acids at WFM during this pollution event. WRF-Chem simulations were performed for the heat wave and pollution event to provide the necessary meteorological and chemical input data to conduct Lagrangian chemical box modeling. BOXMOX was subsequently used for a detailed assessment of

the gas-phase chemistry involved in organic acid production. Gas phase box modeling results are compared to cloud water measurements made at WFM. Additionally, a simple gas + aqueous box model was employed to determine if cloud chemistry contributed to overall organic acid concentrations. Finally, the impacts of anthropogenic emissions on organic acid production will be discussed.



## 2 Description of the Pollution Event

The July 1-2, 2018 pollution event was chosen as case a study to investigate the chemical production of organic acids. This event impacted much of the northeast United States, including WFM, coinciding with a regional heat wave with temperatures reaching 35° C (Figure S1) in several locations. Many locations, particularly the New York City Metropolitan area, saw $O_3$ mixing ratios exceeding National Ambient Air Quality Standards, with mixing ratios reaching over 100 ppbv. (Tian et al., 2020; Tran et al., 2023).

### 2.1 WFM Observations

At WFM, concentrations of several chemical species including organic acids in both cloud water and in the gas phase were considerably greater than normal during this event. Information about cloud water collection protocols at WFM can be found in Lawrence et al. (2023). Briefly, an automated Mohnen omni-directional cloud water collector is used to collect warm cloud water (i.e. $> 0°C$) form non-precipitating clouds between the months of June and September. Samples were collected in a
refrigerated accumulator that dumps into a refrigerated sample bottle every 12 hours. Samples were then analyzed for sulfate ($SO_4^{2-}$), nitrate ($NO_3^-$), ammonium ($NH_4^+$), calcium, ($Ca^{2+}$), magnesium ($Mg^{2+}$), potassium ($K^+$), sodium ($Na^+$), chloride ($Cl^-$), pH, conductivity, water soluble organic carbon (WSOC), and organic acids, including HCOOH, $CH_3COOH$, and OxAc. Organic acids were measured by the Adirondack Watershed Institute using a Lachat QC 8500 Ion Chromatograph, along with $SO_4^{2-}$ and $Cl^-$. A manuscript focusing on the organic acid measurement methods and observations will be submitted separately.
The current work focuses on three of the measured organic acids, HCOOH, $CH_3COOH$ and OxAc, as these are the three most common organic acids found in cloud water at WFM and other locations (Herckes et al., 2013). Trace gases are measured continuously year-round, with chemical species including ozone ($O_3$), oxides of nitrogen (NO, $NO_2$ and $NO_y$), and sulfur dioxide ($SO_2$). More information about the gas phase dataset can be found in (Brandt et al., 2016).

The pollution event consisted of some of the highest concentrations of the season for $SO_4^{2-}$, $NH_4^+$, WSOC, HCOOH,
$CH_3COOH$, and OxAc. (Figure 2), with individual samples of HCOOH and $CH_3COOH$ exhibiting concentrations greater than 100 $\mu$eq $L^{-1}$ and contributing to approximately 30% of measured anions. Additionally, $O_3$ and $NO_y$ mixing ratios were above the 90th percentile of mixing ratios for this event, as compared to the rest of the 2018 summer season (June through September), coinciding with the highest temperatures of the cloud collection season (Figure S2). The relatively high mixing ratios of these trace gases may indicate significant anthropogenic influence. The cloud event focused on two cloud samples
collected between 7/1/2018 0:00 UTC and 7/1/2018 15:00 UTC, with cloud liquid water content (LWC) values reaching up to 1.25 g $m^{-3}$ (Figure S4). The July $1^{st}$ event was chosen for the modeling study as the duration of this cloud event was substantially longer than the event on July $2^{nd}$, making it better suited for modeling.

### 2.2 Determining Total Organic Acid Mixing Ratios from Cloud Water Observations

Currently at WFM, organic acids are measured only within cloud water. However, substantial concentrations of low molecular
weight organic acids have been previously shown to be in the gas phase (Khwaja, 1995). Gas phase and total mixing ratios of

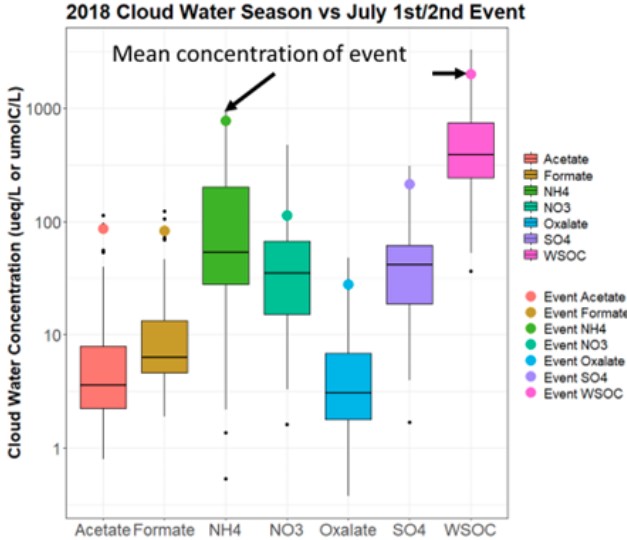

**Figure 2.** Cloud water concentrations of Acetate ($CH_3COOH$), formate (HCOOH), $NH_4^+$, $NO_3^-$, oxalate (OxAc), $SO_4^{2-}$, and WSOC from the 2018 of June-September cloud water season. WSOC is reported in units of $\mu$mol C $L^{-1}$, whereas all other analytes are reported in units of $\mu$eq L$^{-1}$. The 25$^{th}$, 50$^{th}$, 75$^{th}$ percentiles are marked by the colored boxes, the vertical lines represent the 1.5* the inter-quartile range, and the black dots represent values outside the vertical lines.

organic acids can be estimated, assuming the organic acid is in equilibrium with the atmosphere, as a function of the acid's Henry's Law constant, cloud LWC, temperature, pressure, and pH of the cloud droplets using the following equation:

$$\mathrm{OrgAcid_{tot}} = 10^{12}(*\frac{\mathrm{Q_{LWC}(RT)OrgAcid_{aq}}}{\mathrm{P}} + \frac{\mathrm{OrgAcid_{aq}}}{\mathrm{K_{Heff}P_{atm}}}) \tag{1}$$

where $\mathrm{OrgAcid_{tot}}$ is the calculated sum of gas phase and aqueous phase organic acid mixing ratios in pptv, $10^{12}$ is a
conversion factor to convert the mixing ratio to pptv, $Q_{\mathrm{LWC}}$ is the cloud LWC in L m$^{-3}$, R is the universal gas constant (8.314 m$^3$ Pa K$^{-1}$mol$^{-1}$), T is the ambient temperature in K, P is the ambient pressure in Pa, $\mathrm{OrgAcid_{aq}}$ is the concentration of the specific organic acid measured in the cloud water in mol L$^{-1}$, $\mathrm{P}_{atm}$ is the ambient atmospheric pressure in atm, $\mathrm{K_{Heff}}$ is the temperature and pH dependent effective Henry's law constant for the given organic acid in mol atm$^{-1}$. The pH dependency of $\mathrm{K_{Heff}}$ for monocarboxylic acids can be calculated by:

$$\mathrm{K_{Heff}} = \mathrm{K_H}(1 + \frac{\mathrm{K_a}}{[\mathrm{H^+}]}) \tag{2}$$

while for dicarboxylic acids, $\mathrm{K_{Heff}}$ can be calculated by:

$$\mathrm{K_{Heff}} = \mathrm{K_H}(1 + \frac{\mathrm{Ka_1}}{[\mathrm{H^+}]} + \frac{\mathrm{Ka_1Ka_2}}{[\mathrm{H^+}]^2}) \tag{3}$$




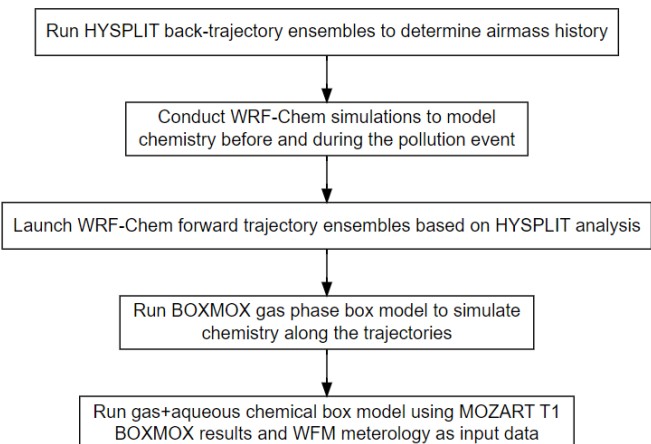

**Figure 3.** Procedure for the modeling analysis organic acids.

where $K_H$ is the standard Henry's law constant of the organic acid, $K_a$ is the acid dissociation constant for monocarboxylic acids, $Ka_1$ and $Ka_2$ are the first and second dissociation constants for dicarboxylic acids and $[H^+]$ is the acidity of the cloud droplets. The temperature dependence of the Henry's Law constant is :

$$K_{Heff} = K_H * \exp(\frac{\Delta H_s}{R} * (\frac{1}{T_2} - \frac{1}{T_1})) \tag{4}$$

where $T_2$ is the ambient temperature, $T_1$ is the reference temperature of 298.15 K, and $\Delta H_s$ is enthalpy of dissolution described in Sander (2023) . The values used for the above calculations can be found in Table S1. $K_H$ values of HCOOH, CH$_3$COOH and OxAc are taken from Sander (2023), while $K_a$ values were taken from Seinfeld and Pandis (2016). The associated pH values of the two cloud samples used in this study are 4.50 and 4.56, while the temperatures are 292.17 K and 292.12 K respectively.

## 3 Modeling Setup

This work uses a combination of modeling techniques, including ensembles of HYbrid Single-Particle Lagrangian Integrated Trajectory (HYSPLIT) back-trajectories (Stein et al., 2015), the WRF-Chem chemical transport model, gas-phase box modeling, and box modeling of gas and aqueous chemistry. This methodology is used to allow for more detailed investigation of the underlying chemistry impacting organic acid formation. It is challenging to investigate chemical processing of an air mass upwind of a location in detail using chemical transport models alone. A Lagrangian approach coupled with a chemical box model allows for the detailed investigation of the underlying chemistry involved in the production of organic acids. Figure 3 summarizes the step by step procedure for this modeling process.



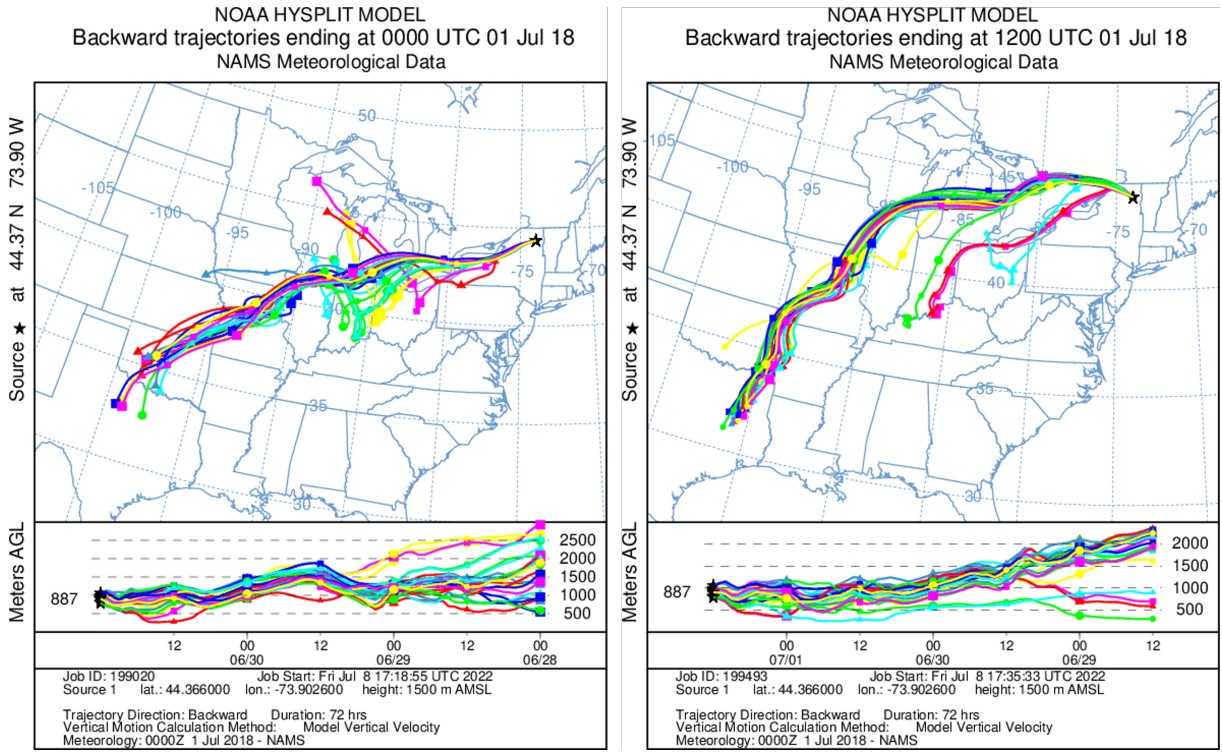

**Figure 4.** HYSPLIT back-trajectory ensembles ending at the summit of WFM (1500m) on July 1st, 2018 at 0:00 UTC and 12:00 UTC. Trajectory ensembles typically flew over Jefferson MO.

### 3.1 HYSPLIT Back Trajectory Analysis

Three-day ensemble back trajectory analysis was conducted to determine the source location of the pollution event using the (HYSPLIT) model (Stein et al., 2015). The receptor site for the trajectories is the summit of WFM, 44.37° N, 73.9° W, 1500 m above sea level. The meteorological data used for these calculations was the North American Mesoscale (NAM) 12kmx12km dataset (more information on the meteorology data can found at https://www.ready.noaa.gov/archives.php). The trajectories consistently flew near the surface in central Missouri near Jefferson City approximately 2 days prior to the pollution event at 145    WFM (Figure 4). This location was therefore chosen to launch the WRF-Chem forward trajectories.

### 3.2 WRF-Chem

#### 3.2.1 Model Run Description

The chemical transport model used for these simulations was the Weather Research and Forecasting Model with Chemistry (WRF-Chem) v4.0.3 (Grell et al., 2005; Fast et al., 2006). Multiphase chemistry including gas, aerosol, clouds, and rain were 150    included within the simulation. A five-day simulation was performed from 6/27/2018 0:00 UTC to 7/2/2018 12:00 UTC with





a 12kmx12km horizontal grid resolution and 43 vertical layers from the surface to 50 hPa. A detailed description of the WRF-Chem simulation parameters and a map of the WRF-Chem domain can be found in section S3 and Figure S3 of the supplemental material.

### 3.2.2 WRF-Chem Evaluation

$O_3$ and PM 2.5 data collected by the EPA's Air Quality System (AQS) monitoring program (US EPA, 2024) were used to evaluate the capabilities of WRF-Chem to represent the pollution event. There is reasonable agreement of surface $O_3$ and $PM_{2.5}$ in the simulations compared to observations collected before and during the pollution event in the Northeast U.S. (Figure 5). The airmass associated with this pollution event was characterized by a combination of high temperatures over the Great Plains region that moved eastward towards the Great Lakes region before reaching the Northeast, under the influence of

a large high pressure system. WRF-Chem properly captured the warm temperatures that moved across the Midwest into the Northeast (Figure S5). There was potential evidence for an influence from wildfire activity from the Southeast U.S. according to the WRF-Chem simulations, but it was unclear if emissions from these fires contributed significantly to the pollution event. To determine potential fire impact, a WRF-Chem simulation that did not include any biomass burning emissions was run for the same time interval as the original simulations. Comparisons of these simulations found virtually no contribution of biomass

burning emissions to $PM_{2.5}$ mass concentrations, $O_3$ mixing ratios, or trace gases important in the formation of organic acids (Figure S6), indicating this pollution event was primarily driven by biogenic and/or anthropogenic emissions.

Three air quality monitoring sites in New York were chosen for time-series evaluations of WRF-Chem, including Pinnacle State Park (PSP) in the Southern Tier of New York, Queens College in New York City, and measurements at both the summit and the old ski lodge below the summit of WFM (Figure S7). More information about the data collected at these sites can

be found in Brandt et al. (2016) and Ninneman et al. (2020). There is strong agreement between model and observations for both temperature and $O_3$ at PSP and Queens College after allowing for 24 hours of model spin up time. However, there is fairly substantial disagreement at WFM for temperature and $O_3$. The complex geography of the Adirondack Mountains are likely not properly captured with a 12kmx12 km horizontal grid resolution, which may cause local meteorological conditions to be modeled improperly, particularly as it relates to the planetary boundary layer. However, WRF-Chem demonstrates skill

in both rural and urban settings with simpler geography, indicating that the simulations are producing realistic conditions of traditionally modeled compounds like $O_3$. $PM_{2.5}$ shows reasonable agreement between model and observations for WFM and PSP, but is greatly overpredicted at Queens College. The causes behind this overprediction remain unclear but are beyond the scope of this work.

### 3.2.3 Forward Ensemble Trajectory Analysis

A feature in WRF-Chem is to monitor air masses through forward trajectories. With an input file, trajectories can be launched at specified latitude-longitude-height locations and times. The trajectory code uses resolved winds (u, v, w) to determine the location of the air mass at each time step. Several variables can be monitored along the trajectory including prognostic and diagnostic information (https://www2.acom.ucar.edu/sites/default/files/documents/Trajectory.desc_.pdf). During the WRF-Chem



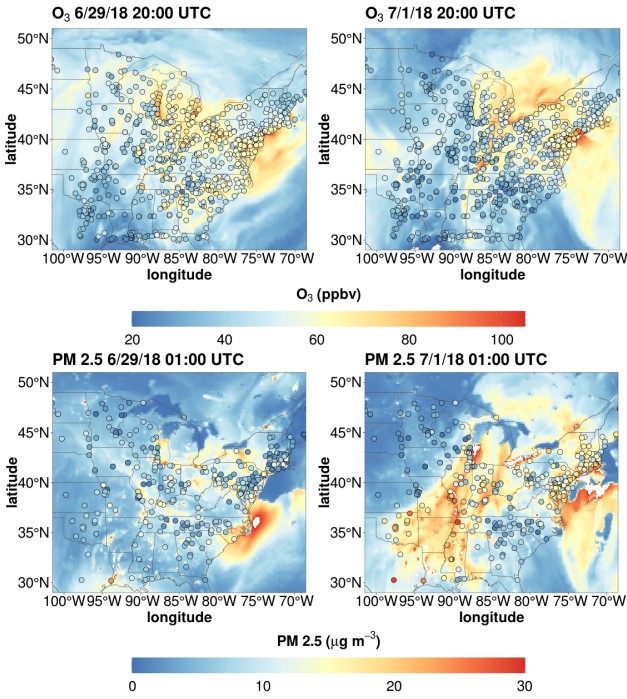

**Figure 5.** WRF-Chem Results for: a) Ozone and b) PM$_{2.5}$ before and during the pollution event that impacted the northeast U.S. Points represent monitoring station observations from the U.S. EPA's AQS monitoring program

simulation, 10 sets of 75 forward trajectories were launched near Jefferson City, Missouri at 38.5° N and 92.5° W. This location was chosen based on the HYSPLIT back trajectory analysis. The starting latitude and longitude of the trajectories was perturbed by +/- 0.1° and +/- 0.2° and were launched at 3 starting heights of 750m , 1000m and 1250m every 2 hours starting at 6/28/2018 22:00 UTC and ending at 6/29/2018 16:00 UTC. To limit analysis to trajectories that influenced WFM, only trajectories that flew within 1° latitude and longitude and below 3000m AGL were considered for chemical box modeling. Of the 750 trajectories launched, 556 trajectories (74.1%) reached WFM.

### 3.2.4 Chemical Box Modeling

The chemical box model, BOXMOX, was used to simulate the gas phase chemistry along the trajectory pathways. BOXMOX uses a Kinetic PreProcessor (KPP) with a Rosenbrock ODE solver (Knote et al., 2015). Necessary box model input parameters were obtained from output data from the WRF-Chem forward trajectories, providing information for initial conditions, emissions (biogenic, anthropogenic, and biomass burning), background conditions, photolysis rate constants, and environmental conditions (temperature, pressure, planetary boundary layer height). Initial conditions are determined by using the mixing ratios at time 0 of the launch locations of the given trajectory. Photolysis rates were provided at a 15 minute time resolution. while emissions, environmental conditions, and background conditions were provided at a 1 hour time resolution. Emissions





were assumed to be zero if the trajectory height was above the top of the boundary layer. In order to account for entrainment of background air into the air parcel, a first order mixing rate constant was set to $1.17 \times 10^{-5}$ s$^{-1}$, associated with a dilution time of approximately 24 hours, consistent with values used in other works (Wolfe et al., 2016; Decker et al., 2019). Sensitivity analysis of this dilution constant in Section S7 reveals that while there were noticeable impacts on organic acid production, the conclusions of this work were not impacted (Figure S8), as will be discussed further in Section 4. Background air is determined by a 60x60km WRF-Chem average mixing ratios of the chemical species of interest at the height of the trajectory.

Two gas phase mechanisms were used for the BOXMOX simulations; the Model for Ozone and Related chemical Tracers version (MOZART) T1 and the Master Chemical Mechanism (MCM) version 3.3.1. Two mechanisms were chosen to determine if a simpler mechanism is sufficient in simulating organic acid chemistry that is more explicitly represented in the more complex mechanism of MCMv 3.3.1. MOZART T1 contains 151 chemical species and 352 gas phase reactions, as described in Emmons et al. (2020). MCM is a highly detailed chemical mechanism containing 142 emitted non-methane VOC species and nearly 17,000 reactions (Jenkin et al., 2015). The MOZART T1 mechanism simplifies the chemistry of larger VOC species by grouping their chemistry into categories of lumped species. These VOCs include BIGALK (alkane species with more than 3 carbons), BIGENE (alkenes with more than 3 carbons) and XYLENES (all XYLENE species and alkyl benzene species but not TOLUENE or BENZENE). However, the individual VOCs that make up these lumped species are directly represented in MCMv 3.3.1 and need to be translated in to realistic atmospheric mixing ratios. Initially, this was done by using whole air sampler VOC data collected by UC Irvine during the KORUS-AQ field campaign to determine what the average fraction of the lumped species was represented by an individual species. However, a sensitivity study using MCMv 3.3.1 was conducted by setting initial conditions and emissions of the lumped species to 0 to determine if they have a significant role in organic acid production (Figure S9). The results showed that there were virtually no differences in organic acid mixing ratios when removing the lumped species from the simulations and therefore the contributions of their chemistry are assumed to be negligible.

### 3.2.5 Gas + Aqueous Chemical Box Model

In addition to the gas phase box modeling, a simplified gas + aqueous box model was introduced to study the effects of aqueous chemistry effects on organic acid concentrations for the analyzed pollution event. Detailed information on the aqueous box model can be found in Li et al. (2017) and Barth et al. (2021). Briefly, the gas + aqueous box model contains a simplified gas phase mechanism with 64 reactants and 168 reactions. Gas-aqueous phase partitioning of low solubility or slow reacting species is controlled by their Henry's Law coefficients while high solubility species (such as HNO$_3$) or fast reacting species (OH, HO$_2$, NO$_3$ radicals) are controlled by the resistance model developed by Schwartz (1986). The aqueous mechanism contains 45 reactions including conversion of sulfur dioxide (SO$_2$) to SO$_4^{2-}$ via hydrogen peroxide (H$_2$O$_2$) and O$_3$, and the oxidation of C1-C3 carbonyls and organic acids via OH the radical.

A limitation of these simulations is that the forward trajectories produced by WRF-Chem contained no cloud LWC, preventing the inclusion of cloud water chemistry along the trajectories, despite the observed cloud event at WFM. Therefore, a set of stationary aqueous box model simulations were run at the summit of WFM. Hourly meteorological measurements at the summit of WFM (including LWC, temperature, and sea-level pressure) were used to constrain these aqueous simulations. A




complication of stationary box models is the need to account for advection of air upwind of a given location. To minimize the potential influence of changing air masses, model runs were limited to 3 hours, with 30 minutes of gas phase only chemistry at the beginning of each simulation, assuming negligible advection and emissions in this timeframe. Three-hour simulations were run each hour from 6/30/2018 12:00 to 7/1/2018 13:00 EST including periods before, during, and after the polluted cloud event at WFM. Initial conditions of gas phase species were provided from hourly averaged mixing ratios from the MOZART T1 BOXMOX results within $1°$ latitude and longitude of WFM. The authors emphasize that while these aqueous modeling methods are highly simplified, the purpose of the aqueous modeling is to determine whether clouds were likely to have had an appreciable impact on organic acid mixing ratios for this pollution event, rather than trying to precisely quantify the impact of cloud chemical processing on organic acid concentrations.

## 4 Gas Phase Box Model Results

### 4.1 Forward Trajectories

There is very little temporal variability in the WRF-Chem trajectory ensembles during the pollution event based on the median trajectory positions for each launch time, consistent with the HYSPLIT back trajectory results (Figure 6a). Median trajectories rather than mean values are used as median values tend to be less sensitive to outliers than mean values (Wilcox, 2012). The ensemble trajectories indicate that many trajectories are within the boundary layer and are influenced by $NO_x$ emissions from the Chicago Metropolitan Area (Figure 6c). The full set of trajectory ensembles can be found in Figure S10. The trajectories largely travel eastward, with little horizontal variation between the trajectories at each launch date, indicating minimal uncertainty in the forward trajectory analysis. Many trajectories experience significant increases of $NO_x$, up to 4 ppbv, as the airmasses advect over the Chicago Metropolitan area, the likely source of anthropogenic influence on the airmass impacting WFM . Some trajectories(particularly those launched from 2018-06-29 10:00 UTC and 12:00 UTC) are also influenced by emissions from Toronto, ON.

Time series of $O_3$ and $NO_x$ for each of the 10 launch dates reveal good model agreement between MOZART T1 and MCMv 3.3.1 results, indicating that the simpler chemistry within MOZART T1 is sufficient in capturing $O_3$ mixing ratios, which vary only slightly (45-60 ppbv) but typically increase as the simulations progress (Figure S11). Many of the trajectories launched from Missouri show enhanced mixing ratios of isoprene, with median mixing ratios of up to 5 ppbv (Figure S12). This is consistent with previous work within the Ozark region of Missouri (Carlton and Baker, 2011; Schwantes et al., 2020), and is exhibited by the WRF-Chem simulations (Figure S13)

### 4.2 Formic and Acetic Acid

#### 4.2.1 HCOOH Production

There is significant net production of HCOOH by both chemical mechanisms (MOZART T1 and MCM) for all of the trajectory launch dates, particularly for trajectories launched on 6/28 22:00 UTC, 6/29 00:00 UTC and 6/29 10:00 UTC, peaking at




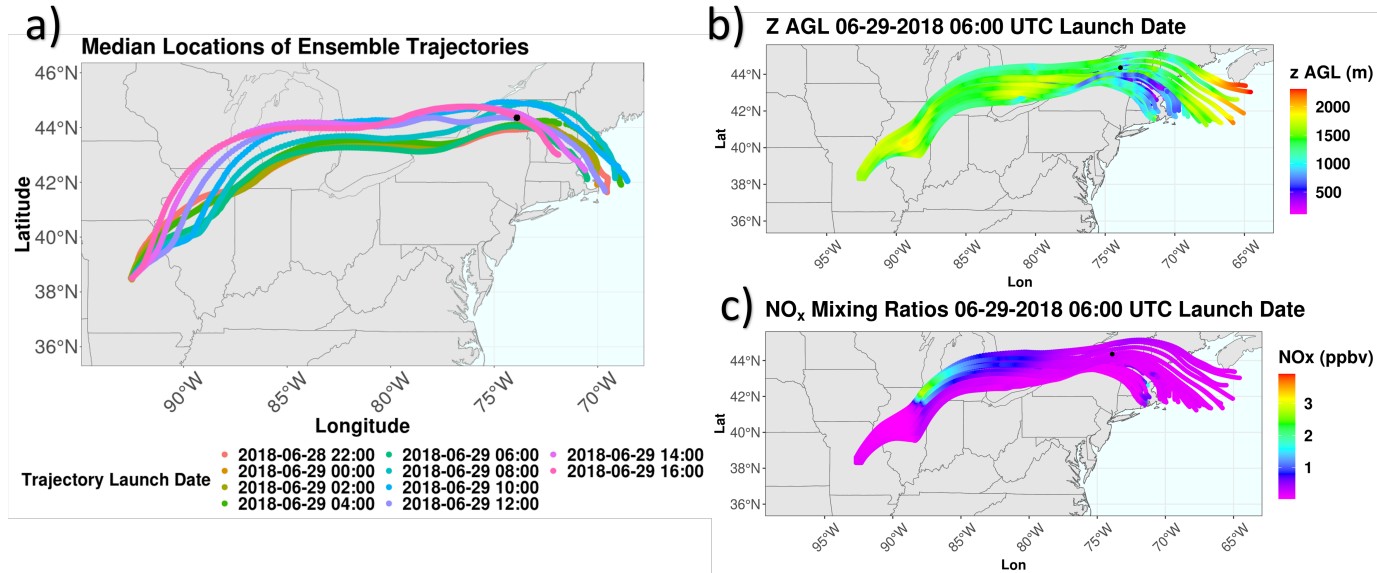

**Figure 6.** a) Median locations of forward trajectory ensembles launched in WRF-Chem, colored by launch date. Forward trajectory ensembles for trajectories launched on 06/29/2018 at 6:00 UTC colored by b) trajectory height above ground level (m), and c) $NO_x$ mixing ratios

mixing ratios of 300 pptv (Figure 7). For all simulations, both mechanisms are in near agreement, with strong production for many sets of trajectories being confined to early in the simulations, before mixing ratios become more controlled by

background conditions as emitted VOC precursors are exhausted. HCOOH for both mechanisms is almost entirely produced by the ozonoylsis of isoprene and isoprene oxidation products, mainly methyl vinyl ketone (MVK) and methacrolein (MACR) (Figure S14). At low mixing ratios of isoprene (< 500 pptv), ethene ($C_2H_4$) becomes the dominant source of HCOOH in MOZART T1, but in these instances, dilution is the major controlling factor. It is worth noting that background mixing ratios of HCOOH are about 5-6 times lower than the peak mixing ratios within the box model simulations, leading to significant

reductions in HCOOH mixing ratios as background air is entrained into the air parcel. The low HCOOH mixing ratios in the background data files are caused by the ozonolysis of isoprene, MVK, and MACR not producing HCOOH within WRF-Chem's MOZART-MOSAIC chemistry mechanism. Using the more comprehensive gas phase chemistry in MOZCART mechanism within WRF-Chem (i.e. MOZART T1 + GOCART aerosol scheme) increases mixing ratios of HCOOH up to 150 pptv (Figure S15). The MOZART-MOSAIC chemistry module was used to simulate aerosol and cloud chemistry for this study to have a

more complete aerosol and cloud chemistry representation that the WRF-Chem T1 chemistry option does not include. Since the background files are extracted from WRF-Chem using the MOZART-MOSAIC, this contributes to a low bias of HCOOH within the box model simulations compared to using the MOZCART mechanism, as discussed in Section 4.2.3.



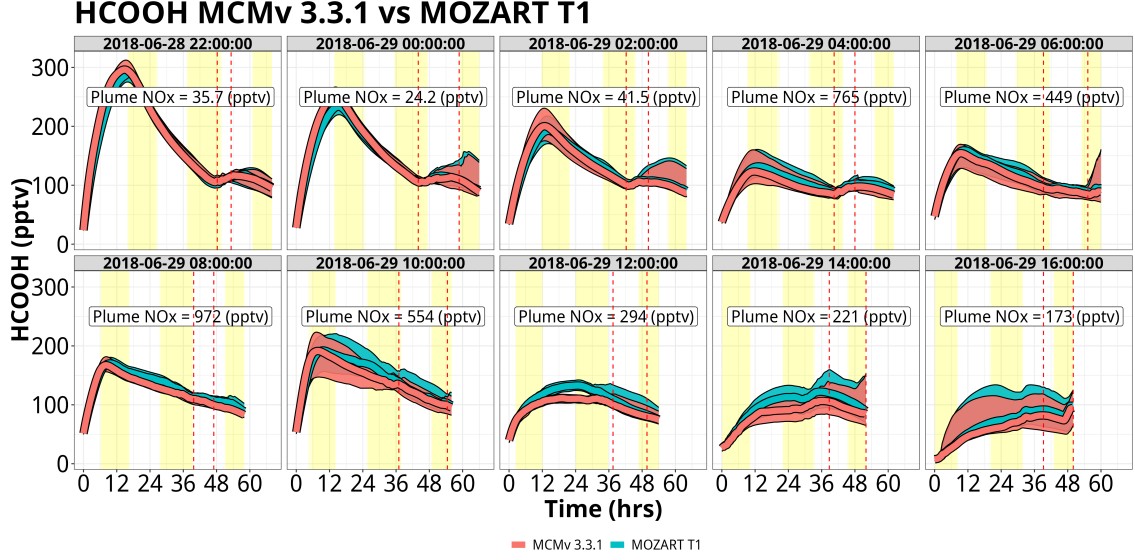

**Figure 7.** Simulation time series of HCOOH mixing ratios for Mozart T1 (blue) and MCM (red) for the WRF-Chem forward trajectory ensembles, separated by launch time. Red and blue lines represent the median value for the ensemble with the shading represents the interquartile range. Yellow shading represents daylight hours. Vertical dashed lines represent the range of times that the trajectories approach WFM. Plum $NO_x$ represents the median $NO_x$ mixing ratios when the trajectories are above the Chicago Metropolitan Area

### 4.2.2 $CH_3COOH$ Production

The mixing ratios of $CH_3COOH$ reach values > 1500 pptv, up to 5 times greater than those of HCOOH (Figure 8). MCM pro-
duces more $CH_3COOH$ than MOZART T1 by up to 500 pptv, with the largest differences occurring within the first few sets of trajectories, i.e. trajectories launched on 6/28 22:00 UTC, 6/29 0:00 UTC and 6/29 2:00 UTC. However, the disagreement be-tween the two chemical mechanisms largely disappears in the later set of trajectories, particularly for the ensembles influenced by higher $NO_x$ mixing ratios (specifically ensembles 6/29 4:00-10:00 UTC) . The major production pathway (greater than 90%) for $CH_3COOH$ is the reaction of the acetyl peroxy radical ($CH_3CO_3$) + the hydroperoxy radical ($HO_2$) or organic peroxy
radicals ($RO_2$). For low $NO_x$ environments, these peroxy radicals can out-compete reactions with NO, leading to significant $CH_3COOH$ production (Figure S16). There are subtle differences in the chemistry between the two mechanisms that contribute to the overall greater production of $CH_3COOH$ in MCM. During the first 20 hours of all sets of trajectories, mixing ratios of $CH_3CO_3$ were approximately the same between the two mechanisms (Figure S17). However, there are important differences in the reactivity of $CH_3CO_3$ within these simulations, particularly as it relates to $RO_2$ radicals. While the overall reactivity of
$CH_3CO_3$ with $RO_2$ radicals is greater in MOZART T1 (as shown in Figure S18), a larger proportion of reactions from $RO_2$ radicals in MCM result in $CH_3COOH$ formation. MCM treats the rate constant and the yield of $CH_3COOH$ from $CH_3CO_3$ + $RO_2$ as the methyl peroxy radical ($CH_3O_2$), while MOZART T1 has only two $RO_2$ species, $CH_3O_2$ and $MCO_3$, that contribute significantly to $CH_3COOH$. Beyond 20 hours, $CH_3CO_3$ mixing ratios are greater in MCM. This is due to stronger production





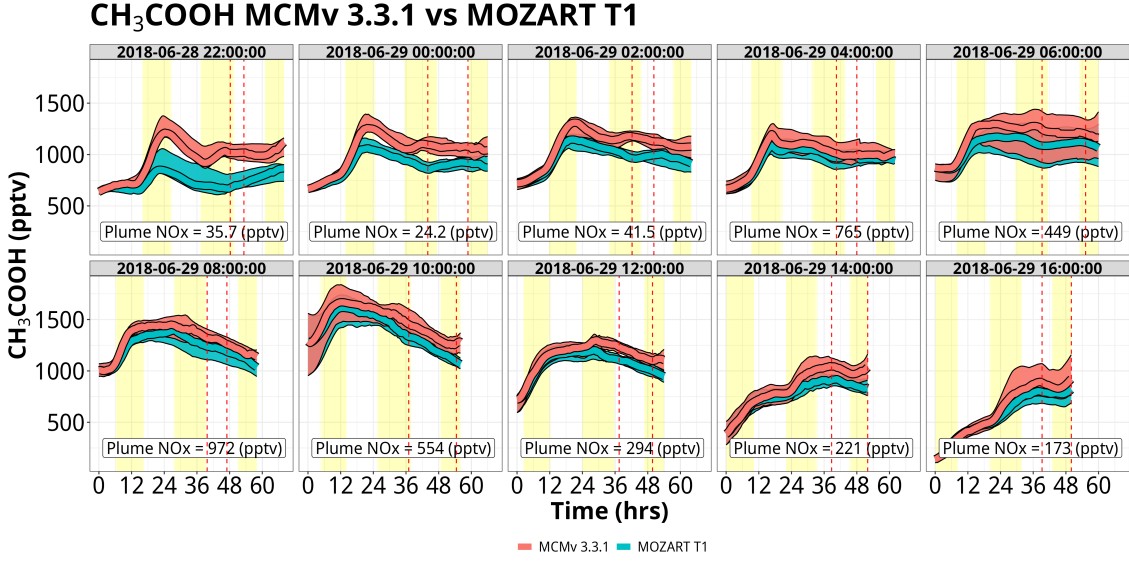

**Figure 8.** Same as Figure 7 but for $CH_3COOH$. Plum $NO_x$ represents the median $NO_x$ mixing ratios when the trajectories are above the Chicago Metropolitan Area

of methylglyoxal within MCM vs MOZART T1, an important precursor for $CH_3CO_3$ from both photolysis and OH (Figure

S19). Disagreements in the rate coefficient for the reaction of OH with peracetic acid also contribute to these discrepancies. Peracetic acid ($CH_3CO_3H$) is not a direct source of $CH_3CO_3$ but rather serves as a chemical reservoir. The $CH_3CO_3H$ + OH rate constant is 3.7x greater in MCM compared to MOZART T1, forcing more $CH_3CO_3H$ to shift back to $CH_3CO_3$ and hence more $CH_3COOH$. There is evidence that this reaction's rate constant is even slower than what is used in either model, indicating that $CH_3CO_3H$ is in reality even more of a permanent sink for $CH_3CO_3$, and thus that both mechanisms may overestimate

$CH_3COOH$ from this pathway (Berasategui et al., 2020).

### 4.2.3 Comparison of gas phase chemistry to cloud water observations

In this section we validate the performance of the gas phase chemical box model by comparing the box model results within $1°$ latitude and longitude of WFM to the derived gas + aqueous phase organic acids (Figure 9). It is assumed that HCOOH and $CH_3COOH$ measured in the cloud water were produced entirely in the gas phase and partitioned into cloud droplets rather

than being produced in the aqueous phase, already existing within the aerosol that the cloud droplets activated on, or being directly emitted. It is also important to note that bulk cloud water may deviate from Henry's Law, even if individual cloud droplets may be in equilibrium with the atmosphere. This can be due to differences in pH of individual cloud droplets, mass transfer limitations (especially for highly soluble or reactive species), and changes in equilibrium due to competing reactions. (Pandis and Seinfeld, 1991; Winiwarter et al., 1994; Wang et al., 2020). Despite these uncertainties, comparing the BOXMOX

results with observations can indicate if the current chemistry represented in the mechanisms can properly model organic acids



in the airmasses arriving at WFM. Average HCOOH mixing ratios increased from 100 pptv to 200 pptv over the course of the simulations, using both mechanisms, while CH$_3$COOH mixing ratios largely remained constant at approximately 1000 pptv. In spite of the substantial disagreements in gas phase production between the two mechanisms, MCM exhibited only 100-200 pptv more CH$_3$COOH than MOZART T1 when it arrived at WFM.

The gas-phase box modeling with both MOZART T1 and MCM substantially underestimated both HCOOH and CH$_3$COOH measured in cloud water by approximately an order of magnitude, implying a significant missing source of organic acids, which may be from gas, particle, or aqueous phases. As mentioned in Section 4.2.1, there is a low bias in the background conditions from the WRF-Chem simulations due to missing ozonolysis reactions of isoprene, MACR, and MVK. However, even the inclusions of the chemistry in the WRF-Chem simulations cannot explain the order of magnitude underestimation of HCOOH

in the BOXMOX results. These results are consistent with other modeling work investigating organic acids, as gas phase box models typically underestimate HCOOH and CH$_3$COOH production, implying that gas phase chemistry alone is not sufficient to properly model these organic acids (Paulot et al., 2011; Millet et al., 2015; Jones et al., 2017). However, it remains unclear the particular reasons for these underestimates. Work by Link et al. (2021) found that ecosystems dominated by isoprene produced greater mixing ratios of organic acids than monoterpene dominated ecosystems, implying that isoprene chemistry

not represented in models might be a missing source of HCOOH and CH$_3$COOH. There is also emerging evidence that cloud droplets may play a unique role in the formation of HCOOH that is not being accounted for in these gas phase simulations. For example, formaldehyde dissolves into cloud droplets, hydrolyzing to form a methanediol, which then partitions back to the gas phase and oxidizes to form HCOOH (Franco et al., 2021). A similar process with other larger aldehydes may be possible, potentially acting as additional sources of larger organic acids.

## 4.3   Comparison of gas + aqueous chemistry to cloud water observations

Cloud chemistry can have profound impacts on organic acid concentrations distinct from gas phase chemistry alone. This section examines the impacts of aqueous chemistry by investigating both total mixing ratios (left) and aqueous concentrations (right) of HCOOH and CH$_3$COOH (Figure 10). The total mixing ratios are useful to show the overall change in organic acid concentrations resulting from chemistry in both phases while the aqueous phase concentrations can be used to directly compare

to cloud water measurements. Despite large concentrations of CH$_3$COOH in the aqueous phase, there is very little change in total mixing ratios of CH$_3$COOH throughout these simulations, indicating a limited role of chemistry (within the gas or aqueous phase) on the overall CH$_3$COOH produced within these simulations. However, HCOOH is almost completely depleted within the aqueous phase, driven largely by the ionic HCOO$^-$ reacting with aqueous phase OH radical, with limited aqueous production from HCHO+OH unable to replace HCOOH. The majority of HCOOH depletion occurs from photochemistry

during the daytime hours of 15-21 and 30-39. Both HCOOH and CH$_3$COOH are greatly underestimated compared to cloud water measurements, similar to the gas phase only results. Model/observational discrepancies are also made worse by the aqueous depletion of HCOOH, suggesting an even greater missing source of gas phase HCOOH, unrepresented aqueous or heterogeneous HCOOH production pathways, or some combination of these processes. The depletion of HCOOH deviates from a previous cloud chemistry modeling study at WFM (Barth et al., 2021). The same aqueous chemical mechanism found



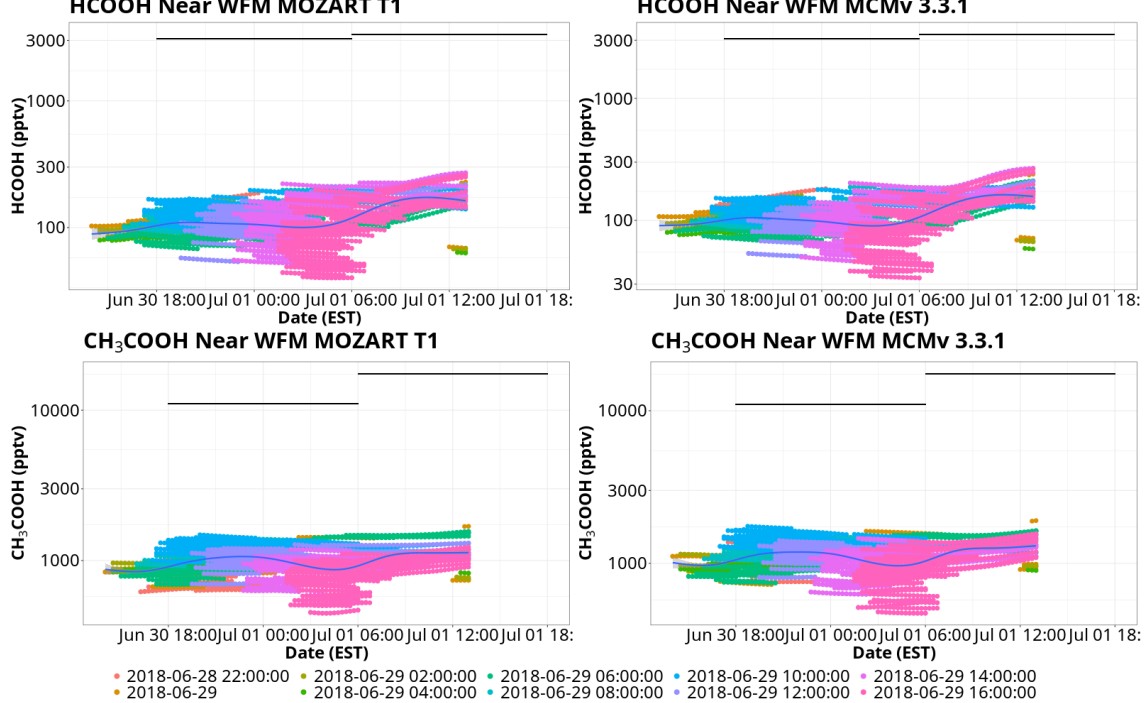

**Figure 9.** Comparisons of model and observational mixing ratios of HCOOH and CH$_3$COOH for MOZART T1 and MCM. Points represent modeled mixing ratios from the trajectory ensembles within $1°$ of WFM, colored by trajectory launch date. Black lines represent the total (gas + aqueous) mixing ratio estimates derived from 12 hour bulk cloud water samples collected at the summit of WFM.

strong production of HCOOH within cloud water, while a more complex aqueous mechanism, CAPRAM $4.0\alpha$ exhibited even stronger production due to reactions involving the aqueous oxidation of CH$_3$CO$_3$H not included in the model used in this study. The reasons for HCOOH depletion in this modeling study remains unclear, but likely is related to missing reactions in one or both of the gas and aqueous phases that are beyond the scope of this work.

## 5  Oxalic Acid

Neither MOZART T1 or MCM produce OxAc despite its known prevalence, as there is no known gas phase chemistry that produces OxAc. Current research points to aqueous chemistry being its dominant source, with glyoxal serving as an important precursor (Sorooshian et al., 2006; Lee et al., 2011). Since glyoxal serves as an important precursor gas for organic acid production, it is worth investigating the gas-phase chemistry controlling glyoxal production.





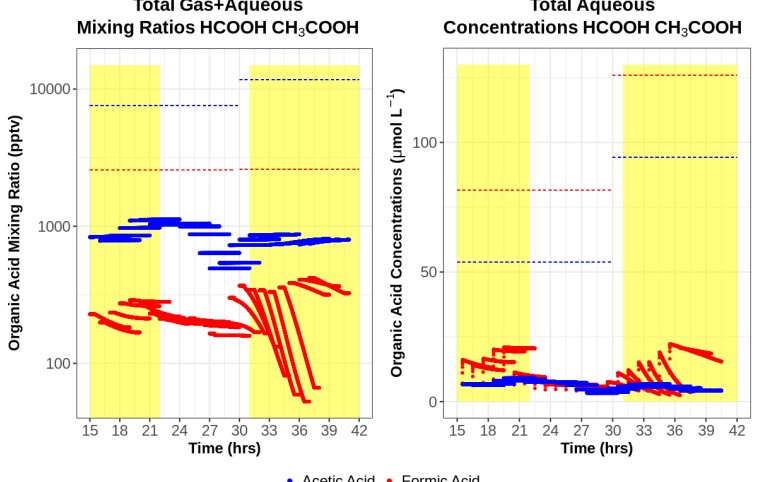

**Figure 10.** Total (gas + aqueous) mixing ratios (left) and aqueous phase concentrations (right) of HCOOH (red) and CH$_3$COOH (blue) from the simple gas + aqueous box model run at the summit of WFM during a cloud event that occurred from June 30$^{th}$ to July 1$^{st}$, 2018. Dashed horizontal lines represent cloud water concentrations measured at WFM during this period. Total mixing ratios in the left plot were derived from the cloud water measurements using Eqs. 1-3.

## 5.1 Glyoxal Production

Glyoxal shows complex differences between the two gas phase mechanisms (Figure11). In the first two sets of trajectories, MCM produces up to 2x more glyoxal than MOZART T1 but for later sets of trajectories, such as 6-29-2018 at 8:00 and 10:00 UTC, MOZART T1 produces up to 50 pptv more glyoxal than MCM. The higher glyoxal mixing ratios within MOZART T1 are associated with higher daytime isoprene mixing ratios (greater than 1 ppbv) coupled with higher NO$_x$ mixing ratios over the Chicago Metropolitan area. Further investigation of the major chemical production pathways between the two mechanisms

reveals that MCM predicts considerable ozonolysis chemistry of isoprene oxidation products (including a strong source from the ozonolysis of a hydroperoxy aldehyde or C5HPALD2 in MCMv3.3.1), a source that is not included in MOZART T1 (shown in Figure S20). Trajectories launched on 2018-06-28 22:00 UTC show the strongest nocturnal production within MCMv 3.3.1 as the simulation starts towards the end of the day. Photochemistry only has a few hours to oxidize nearly 5 ppbv of isoprene, and as a results only produces typically short-lived second-generation oxidation products such as C5HPALD2 (with a chemical

lifetime of 1 hour when OH = 5x10$^6$ molecules cm$^{-3}$ s$^{-1}$), which then strongly reacts with O$_3$ at night to form glyoxal.

In trajectories influenced by anthropogenic NO$_x$, such as ensembles launched on 6/29/18 6:00 UTC and 10:00 UTC, a major glyoxal production pathway in MOZART T1 is the reaction of a lumped peroxy radical (XO$_2$) with NO, where XO$_2$ is a lumped species representing peroxy radicals formed in the oxidation of isoprene by-products including isoprene epoxydiol (IEPOX), hydroperoxyaldehyde (HPALD), and an a unsaturated hydroxyhydroperoxide (ISOPOOH), and represents the day-

time chemistry that leads to greater glyoxal production in MOZART T1 compared to MCMv 3.3.1. Similar to CH$_3$COOH, the





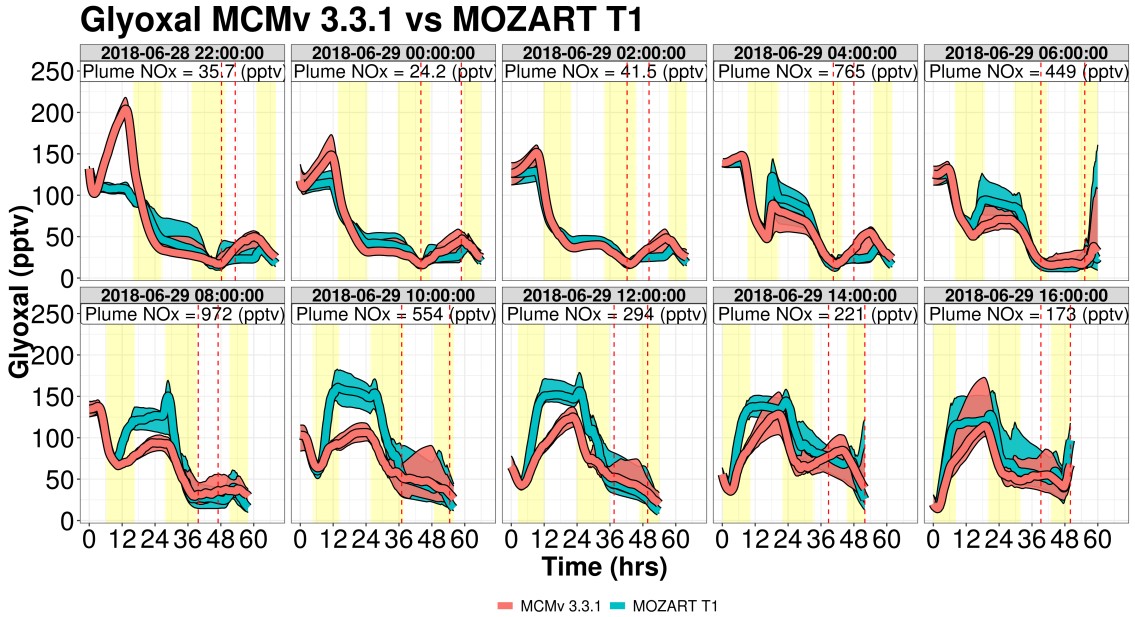

**Figure 11.** Same as Figure 7 but for glyoxal

disagreements between the two mechanisms largely disappear for glyoxal as trajectories arrive at WFM, as primary VOCs are depleted and glyoxal is oxidized or the air parcel entrains background air (Figure S21).

## 5.2 Oxalic acid cloud chemistry

Results of the gas + aqueous modeling find substantial aqueous phase production of OxAc that corresponds with a sharp
aqueous phase depletion of glyoxal (Figure 12). OxAc production is confined to the daytime, as the OH radical is the major driver of OxAc production chemistry within the model. The concentrations of OxAc are well within an order of magnitude of measured cloud water concentrations. These simulations suggest that aqueous chemistry of small carbonyl compounds such as glyoxal can largely explain the observed concentration of organic acids such as OxAc. It is important to note that this is a simplified aqueous box model that focuses on 2 or 3 carbon organic acid chemistry that is better suited for chemical transport
models. There are aqueous chemical mechanisms that contain larger organic compounds and more aqueous phase reactions that likely better capture the chemical complexity in cloud droplets and wet aerosol (McNeill et al., 2012; Mouchel-Vallon et al., 2017; Bräuer et al., 2019). Additionally, other types of chemistry such a transition metal ion chemistry (Zuo and Hoigne, 1992; Sorooshian et al., 2013) or reactions involving organic nitrogen or organic sulfur compounds (Pratt et al., 2013; Lim et al., 2016) are not included in this mechanism that could have direct or indirect impacts on organic acid formation. Uncertainties
of Henry's Law for OxAc and precursor gases may also contribute to uncertainties in overall OxAc production. Despite these uncertainties, the model results provide strong evidence that under atmospherically relevant conditions, aqueous chemistry can have major impacts on concentrations of organic acids like OxAc and HCOOH.



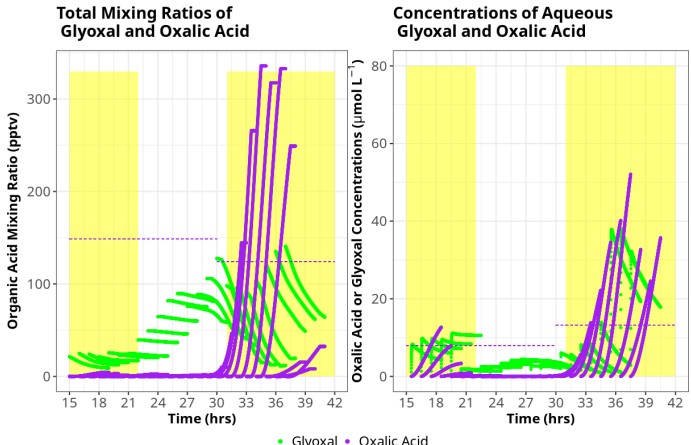

**Figure 12.** Same as Figure 10 but for glyoxal (green) and OxAc (purple). Dashed horizontal lines represent observations from WFM cloud water.

## 6 Discussion

### 6.1 Influence of anthropogenic NO$_x$ emissions on organic acid formation

Strong isoprene emissions from Missouri are a major contributor to all three organic acids discussed in this work. However, several air parcels modeled in this study are also influenced by anthropogenic NO$_x$ emissions from the Chicago metropolitan area, which impacted the oxidation pathway of isoprene in these simulations. A high NO$_x$ versus low NO$_x$ chemical regime for specific VOCs is often defined by whether RO$_2$ predominately reacts with NO or HO$_2$, which can change the overall oxidation pathway of the VOC. The [NO]/[HO$_2$] ratio can serve as a useful proxy for the NO$_x$ regime to explore the impacts of

anthropogenic NO$_x$ on organic acid production. The impact of NO$_x$ emissions from the Chicago Metropolitan area on HCOOH production are subtle, as the dominant production pathway of HCOOH is from isoprene ozonolysis. NO$_x$, coupled with warm temperatures, is directly related to O$_3$ production and high NO$_x$ could therefore contribute indirectly to HCOOH formation. However there is little correlation between [NO]/[HO$_2$] ratios with HCOOH production rates in these simulations (Figure S22), as the vast majority of HCOOH production in all trajectory ensembles occurred during the first 10-15 hours of the simulation,

before trajectories reached the Chicago Metropolitan Area and the primary VOCs responsible for HCOOH production (mainly isoprene) are largely exhausted. NO$_x$ emissions have a more direct impact on CH$_3$COOH production, particularly within MCM, with the production of CH$_3$COOH being reduced by up to 3x for [NO]/[HO$_2$] ratios greater than 10 (Figure S23). This reduction is caused by NO out-competing HO$_2$ and RO$_2$ to react with CH$_3$CO$_3$ due to elevated anthropogenic NO$_x$ emissions from the Chicago Metropolitan Area, thus reducing the major production pathway of CH$_3$COOH.

It is not possible to directly investigate the role of anthropogenic NO$_x$ on OxAc using these simulations, as there is no gas phase production of OxAc in either mechanism. Instead, glyoxal's NO$_x$ dependency can be examined as a proxy for OxAc. Both gas phase mechanisms show glyoxal production increasing with [NO]/[HO$_2$] ratios, with a stronger relationship within



MOZART T1 simulations, due to the parameterized $XO_2$ +NO reaction (Figure S24). The timing of the $NO_x$ emissions is as important as the strength of the emission sources as it relates to glyoxal. The trajectory ensemble launched on 06/29/2018 8:00 UTC exhibited some of the highest $NO_x$ mixing ratios in the simulations (Figure S11), but these emissions arrived mostly at night, muting the impact they could have on glyoxal production. Compare this to the trajectories launched on 06/29/2018 10:00 UTC, where anthropogenic $NO_x$ contributes to a glyoxal production rate 2 times greater than the 06/29/2018 8:00 UTC trajectories in the first 40 hours of the simulations, despite $NO_x$ mixing ratios being approximately 2 times smaller. These results indicate daytime anthropogenic influence increased overall glyoxal production and its likely oxidation products such as oxalic acid, but this influence was decreased due to the timing of the $NO_x$ emissions.

## 6.2   Modeling Uncertainties

There are several processes that may contribute to uncertainties in modeling organic acids that arise from unknowns in both gas-phase and aqueous-phase chemistry as well as the lack of measurements of a suite of trace gases and aerosol composition and concentrations. There are large disagreements between MOZART T1 and MCMv 3.3.1 in the production of $CH_3COOH$ and glyoxal. While there is mechanism agreement as trajectories arrive at WFM, this agreement is caused by entrainment of background air controlling the $CH_3COOH$ and glyoxal mixing ratios rather than similar chemical production rates. Investigating the production of these gases in another location or on a different date would likely lead to different results. While changing the entrainment parameter within the box modeling did not impact the conclusions of this work, changes in this parameter did have an appreciable impact on the magnitude of the organic acid mixing ratios, and thus increasing the uncertainty in modeling organic acid production. The model runs underestimating HCOOH and $CH_3COOH$ by an order of magnitude imply missing chemistry, but it is unclear if this is due to gas and/or aqueous chemistry.

While the gas+aqueous chemistry model produces measured OxAC concentrations, the model is missing known processes that could serve as OxAc sources such as the oxidation of larger organic compounds (Tilgner and Herrmann, 2010; Barth et al., 2021), sinks such as iron-oxalate complexes (Zuo and Hoigne, 1992; Sorooshian et al., 2013; Mouchel-Vallon et al., 2017), or key controls of the oxidant budget like photo-fenton reactions (Deguillaume et al., 2005; Nguyen et al., 2013))

The box model simulations also lack representation of organic aerosol that may contribute further uncertain. Organic acids may have already existed within aerosol before cloud formation, providing a direct source of organic acids to cloud water before any chemistry has occurred. Carbonyl compounds have also been detected within aerosol samples (Liu et al., 2022; Wang et al., 2022) , which can be then oxidized after cloud droplet activation to form organic acids. WSOC can serve as an important sink for aqueous-phase OH, which can either enhance or reduce organic acid production depending on the amount of organic acid precursors available for reaction (Arakaki et al., 2013; Tilgner and Herrmann, 2018).

In addition to uncertainties of modeling components, the lack of field observations of both organic acids and their precursors reduces our ability to constrain organic acid production. Regular monitoring of organic acids and their precursor gases are rare in the Northeast US or elsewhere. VOCs are monitored in networks like the EPA's Photochemical Assessment Monitoring Stations (PAMS), but are designed to assess $O_3$ production, and are therefore constrained to more populated regions . Whiteface





Mountain is the only site in the region that monitors organic acids and there are no recent gas phase organic acid measurements in the region, with the most recent known measurements occurring in 1991 (Khwaja, 1995).

### 6.3 Future Work

Future work will investigate the impacts of cloud water chemistry on organic acid production in more detail. Specific attention

will be paid to the aqueous phase depletion of HCOOH and why this result differs from a another WFM case study using the same mechanism (Barth et al., 2021). In addition to a more detailed look at the key chemical reactions (ie sinks, sources, oxidant budgets) within the simple gas+aqueous phase mechanism, the aqueous chemistry will be expanded to include key processes that were not represented in this work, including metal-organic complexes and associated photo-chemistry, photo-fenton chemistry, and the inclusion of larger organic compounds in the mechanisms. This updated chemistry will then be compared

to observations to see if the improved mechanism can better describe HCOOH, $CH_3COOH$, and OxAc concentrations.

### 7   Summary and Conclusions

This study used a combination of WRF-Chem and Lagrangian chemical box modeling to investigate the major chemical processes that impact organic acid formation in both the gas and aqueous phases at Whiteface Mountain, NY (WFM) during a pollution event on July 1, 2018 that led to record high organic acid concentrations. HYSPLIT ensemble back-trajectory

analysis determined that WFM received influence from Central Missouri, a region with strong biogenic VOC emissions, and anthropogenic emissions from Chicago Metropolitan Area. WRF-Chem simulations were used to simulate before and during the pollution, and to launch forward trajectories based on the HYSPLIT results. WRF-Chem was then used to provide input necessary for chemical box modeling along the trajectories. To determine if gas-phase chemistry can explain the organic acid concentrations measured at WFM, the box model, BOXMOX, was run with two gas phase mechanisms (the Model for OZone

and Related Tracers or MOZART T1 and the Master Chemical Mechanism or MCMv3.3.1) The MOZART T1 mechanism is a condensed gas-phase mechanism while MCMv 3.3.1 is more detailed, allowing evaluation of whether MOZART T1 can sufficiently predict organic acid production compared to MCMv 3.3.1. The gas phase box model results were then used as input for a simple gas+aqueous box model run at the summit of WFM to investigate the potential role of aqueous chemistry on organic acids. Strong biogenic emissions of isoprene from Missouri driven by a heat wave were responsible for the strong

production of organic acids, with influence from anthropogenic inputs of $NO_x$ from the Chicago metropolitan area.

    The two gas phase mechanisms used in the BOXMOX simulations showed good agreement in HCOOH production, with ozonolysis chemistry from isoprene, MACR, and MVK serving as the major sources. MCMv 3.3.1 produced up to 40% more $CH_3COOH$ than MOZART T1 under high isoprene but low $NO_x$ conditions due a stronger $CH_3CO_3$+ $HO_2$ chemical pathway. The two gas phase mechanisms differed in their calculation of glyoxal production. MCMv3.3.1 produced more glyoxal from

the nocturnal ozonolysis of hydroperoxy aldehyde or C5HPALD2, a low $NO_x$ oxidation product of isoprene, while MOZART T1 produced more glyoxal under higher $NO_x$ conditions where NO + $XO_2$ dominated. The disagreements between the two mechanisms for $CH_3COOH$ and glyoxal largely disappear as they arrive at WFM, but this is due the entrainment of back-





ground air dominating mixing ratios after emitted primary VOCs have been exhausted. Both gas phase mechanisms greatly underpredicted HCOOH and CH$_3$COOH by an order of magnitude in comparison to measurements made a WFM.

To learn how aqueous-phase chemistry could contribute to organic acid formation, a cloud chemistry box model was applied using a simple aqueous-phase mechanism. The gas+aqueous phase box model shows little change in CH$_3$COOH mixing ratios due to aqueous chemistry but exhibits a significant depletion of HCOOH, exacerbating the gas phase underpredictions of HCOOH. Glyoxal mixing ratios showed strong disagreements between upwind of WFM, with MCMv 3.3.1 producing large amount of glyoxal at nighttime from the ozonolysis of an isoprene hydroperoxy aldehyde (C5HPALD2), while MOZART T1

showed stronger production during the day from lumped isoprene oxidation peroxy radical XO$_2$ reaction with NO. Anthropogenic NO$_x$ emissions led to increased glyoxal production in both mechanisms, but the effect was stronger within MOZART T1. There is strong aqueous production of OxAc from carbonyl compounds like glyoxal, with concentrations well within an order of magnitude of cloud water measurements at WFM. The gas+aqueous box modeling indicates that aqueous processing can impact organic acid concentrations.

These results contribute to a growing body of work showing the large uncertainties in modeling organic acids. Furthermore, only a limited number of modeling studies have looked explicitly at OxAc, (Crahan et al., 2004; Ervens et al., 2004; Warneck, 2005; Chen et al., 2007; Myriokefalitakis et al., 2011; Zhu et al., 2019; Barth et al., 2021; Myriokefalitakis et al., 2022) , despite its role as a significant component of SOA mass. A major reason for this is that most chemical transport models contain either no or a crude representation of organic chemistry within cloud droplets. The lack of representation of aqueous organic

chemistry risks the model developing a "clear sky bias", phrase introduced in Christiansen et al. (2020), preventing proper characterization of the chemical properties of organic aerosol. Mechanism development and deployment within these models is necessary to better simulate organic acids concentrations and organic aerosol.

Another contributing factor to uncertainties in organic acid production is the lack of observational data, particularly organic acids in both the gas and aqueous phases. Regular observational studies over a broader range of geographical and tempo-

ral scales are required to better constrain organic acids. VOC measurements of key organic acid precursors like isoprene, methacrolein, methyl vinyl ketone, and glyoxal, especially in regions of high BVOC emissions, are required to better constrain organic acid production. Cloud water chemistry measurements need to be expanded beyond organic acids to include key aqueous precursor gases such as glyoxal and methylglyoxal. Simultaneous gas and aqueous phase field measurements are also necessary, as cloud water measurements alone are not sufficient to properly investigate cloud water processing of organic

carbon. Finally modeling work at different temporal and geographic scales coupled with field observations is necessary for advances in model development and to better understand the processes governing atmospheric chemistry. The procedure of back-trajectory analysis that then initialize forward trajectory runs within WRF-Chem (or another chemical transport model) could be automated to provide insight to researchers during field campaigns and guide laboratory analysis of collected samples to target specific chemical species or processes.

The Northeast US is a region undergoing a significant shift in condensed phase chemical composition from a SO$_4^{2-}$ and NO$_3^-$ dominated system to an organic carbon dominated system, with organic acids representing a larger fraction of total ions in cloud and rain water (Lawrence et al., 2023) . Because of the trend towards a higher fraction of organic acids in cloud water,



it is critical to better understand their production. As the world decarbonizes and anthropogenic emissions of $SO_2$ and $NO_x$ decrease, field campaigns and modeling efforts targeting the Northeast US can serve as a blueprint for other regions of the
world that are experiencing similar changes in atmospheric composition and chemistry, improving the representation in air quality and climate models of aerosol and precipitation composition and therefore inform policy decisions.

*Author contributions.* E. Yerger and D. Kelting conducted the cloud water chemical analysis. P. Casson, R.Brandt, C.Lawrence and S.Lance conducted the cloud water sampling in 2018. M. Barth and C.Lawrence conducted WRF-Chem simulations. C.Lawrence conducted box modeling simulations. J.Orlando consulted on box model results. C.Lawrence and M.Barth wrote the manuscript with contributions from all
co-authors.

*Competing interests.* The authors have the following competing interests: At least one of the (co-)authors is a member of the editorial board of Atmospheric Chemistry and Physics.

*Acknowledgements.* The NSF National Center for Atmospheric Research (NCAR) is a major facility sponsored by the U.S. National Science Foundation under Cooperative Agreement No. 1852977. The authors would like to thank the NCAR Advanced Studies Program Graduate
Visitor Program and NCAR's Atmospheric Chemistry Observations and Modeling Laboratory for providing travel funding to allow in person collaboration between authors C. Lawrence, M. Barth, and J.Orlando. We would like to acknowledge the high-performance computing support from Cheyenne (doi:10.5065/D6RX99HX) provided by NSF NCAR's Computational and Information Systems Laboratory. We acknowledge use of the WRF-Chem preprocessors tool (mozbc, fire_emiss, biogenic emissions, anthropogenic emissions), provided by NSF NCAR/ACOM. C.Lawrence's graduate research is funded by NASA's Future Investigators in NASA Earth and Space Science and
Technology (FINESST) program (award number 80NSSC21K1633) and the NSF's Faculty Early Career Development (CAREER) grant (award number 1945563). Cloud water measurements reported in this paper were supported by the New York State Energy Research and Development Authority (NYSERDA) Contract 124461 (2018-2021). We would also like to thank James Schwab for providing trace gas data from Whiteface Mountain, Pinnacle State Park, and Queens College. WFM trace gas and meteorological measurements were supported by NYSERDA Contract 48971. NYSERDA has not reviewed the information contained herein, and the opinions expressed in this report do not
necessarily reflect those of NYSERDA or the State of New York.



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
