# Peer review of "Process Analysis of Elevated Concentrations of Organic Acids at Whiteface Mountain, New York"

_EGUsphere, 2024_

## Referee Comment (RC2)

The manuscript titled "Process Analysis of Elevated Concentrations of Organic Acids at Whiteface Mountain, New York" by Lawrence et al. investigates the potential role of aqueous chemistry on organic acids using a combination of modeling and observations from Whiteface Mountain. The paper has potential to increase understanding on organic acid formation, however, there are a few concerns that should be addressed prior to publication.

**General Comments**

The manuscript lacks a clear explanation regarding the impacts of its results and how it advances knowledge. In the abstract and introduction, there is discussion of the uncertainties associated with organic acids including how organic acids are commonly not included in detail in models and that few studies on organic acids in the Northeast U.S. exist. The manuscript then states "To address these shortcomings, this study investigates…". How are these shortcomings addressed with the results of this study? This is not clearly explained or highlighted since the conclusions seem to just restate the motivation of the study – there are uncertainties in organic acid production and further modeling and observational studies are needed. While future work can still be needed, it is important to address the significance of the results of this study that was performed and in what particular aspects it furthers current knowledge.

The manuscript tends to use qualitative descriptive language and would benefit from more quantitative evaluations. For example, throughout the manuscript there are statements of "good model agreement", "significant reductions",  "stronger production", "very little change", "substantially underestimated", "little correlation". How are these terms defined? What quantitative values do they represent? A statement such as "reduced by 80%" or "reduced with statistical significance by t-test" is more informative and stronger than "significant reductions". More quantitative results may also help in addressing the above concern.

**Specific/Technical Comments**

Line 7: remove "analysi".

Line 89: "form" should be "from".

Line 251: Remove space before period.

Line 251: Add space after "trajectories".

Line 258: Add period at end of sentence.

Line 331-332: Remove "(left)" and "(right)". This is appropriate for the figure caption and seems unnecessary in the main body text.

Line 340: What do these hour numbers represent? Please clarify.

Figure 1: Please make font size larger. Even with zooming in, some words are too small to be legible.

Figure 5: The caption references a) and b) but the images are not labeled as such.

Figures 7 & 8: In the caption, "Plum" should be "Plume".

Figure 9: What does the blue solid line represent?

Supplement: Section S3 comes before S2? Line 152 references Section S3 for WRF Chem description but it is listed as S2 in the supplement. "Figures S8" under Section S6 should be Figure S7. In Section S7, "Figure S9" should be Figure S8.

---

## Author Comment (AC1)

**Response to Reviewer 1**

We would like to thank the reviewer for their detailed edits and comments that have improved this manuscript. Below you will find our response to each comment/question. The reviewer's comments/questions are in bold italic font.

***In the Abstract the authors state, "aqueous chemistry exacerbated the discrepancies of HCOOH leading to a net depletion". Which discrepancies – between models or in relation to measurements at WFM?***

This sentence was edited to clarify which discrepancies we were referring to. This sentence was changed to:

"The addition of aqueous chemistry exacerbated the discrepancies of HCOOH with observations by leading to a net depletion within cloud water."

***The cloud samples are collected in bulk at 12 hour intervals. Can nighttime vs. daytime differences in the cloud samples be teased out to understand potential accuracy of model differences in their prediction of the predominance of glyoxal production pathway (i.e., nighttime from the ozonolysis of an isoprene hydroperoxy aldehyde vs. daytime oxidation of the lumped peroxy radical XO2 reaction with NO)? Perhaps the NOx emission timing and concentrations are too uncertain.***

This is an interesting question certainly worthy of investigation. However, most of the chemical production of glyoxal occurs upwind of WFM (see Figure S23) before subsequently being controlled by entrainment of background air. Therefore, differences in nighttime vs daytime oxalic acid concentrations measured at WFM are not necessarily indicative of associated nighttime vs daytime chemistry impacting oxalic acid concentrations. Due to the reason stated above, we do not believe that it is possible to test nighttime vs daytime model disagreement.

***The authors assume gas-to-droplet partitioning and do not consider aqueous phase production and then find that the gas-phase predictions are an order of magnitude lower than WFM cloud water measurements. Is there any understanding from Barth et al., (2021) cited in the manuscript as to which process produces more of the carboxylic acids in cloud water? Is it gas-to-droplet partitioning or in-cloud production?***

To evaluate whether gas-phase only chemistry can produce the measured organic acid concentrations, gas-to-droplet partitioning was applied to the measured aqueous-phase concentrations to estimate the gas-phase mixing ratios for HCOOH, $CH_3COOH$, and oxalic acid. In section 4.3, we then considered gas + aqueous-phase chemistry in the Barth cloud chemistry model that represents both gas-to-droplet partitioning and chemistry. The results from these runs

show that gas-to-droplet partitioning was 100x and 10,000x greater for HCOOH and CH$_3$COOH, respectively, than their aqueous-phase chemical reaction sources. Oxalic acid has no gas phase production chemistry, so any production will only have occurred in the aqueous phase. In this current study, cloud water chemistry depletes HCOOH concentrations while the Barth model in the Barth et al. (2021) paper, cloud water chemistry is the major source of HCOOH in cloud droplets, evidenced by strong increases in HCOOH in the aqueous phase while there were no increases in HCOOH during the gas phase only simulations. These differences highlight that cloud water can be either a net sink or net source of HCOOH in different scenarios. A few sentences were included that summarizes the above discussion which read:

"These model results imply that gas-to-droplet partitioning is the major source of HCOOH and CH$_3$COOH in cloud water rather than chemical production within cloud droplets. This is confirmed by comparing the rate of gas-to-droplet partitioning to aqueous production, which is 100x and 10,000x greater for HCOOH and CH$_3$COOH respectively."

And:

"The differences in model results on different dates imply that cloud water chemistry can either be a net source or net sink of HCOOH depending on the given scenario."

***In this paper, the description of model performance is generally subjective and colloquial. Phrases such as "strong agreement", "fairly substantial disagreement" etc. are common. It would be better to provide quantitative assessment. For example, starting on Line 154, the authors state, "There is reasonable agreement of surface O3 and PM2.5 …" Typically models are evaluated with a quantitative description of statistics such as normalized mean bias. Can the authors provide this assessment for PM2.5 and O3 in this simulation and provide context for model performance? I find it is difficult to tell in some areas (e.g., near Lake Michigan) what "reasonable agreement" is. Also, how is the bias/error in the general vicinity of the HYSPLIT trajectory 'upstream' of WFM, specifically?***

The WRF-Chem Evaluation section has been updated to be more quantitative in reporting model performance. Pearson correlation coefficients and Mean Bias Error (MBE) maps have been added to the supplemental material (Figures S7 and S8), and these same statistics have been added to Figure S9. Additionally, the text was updated to include these model performance statistics and how the model performs along the HYSPLIT trajectories. The section was updated to read as follows:

"Modeled O$_3$ exhibited a strong positive linear correlation (r > 0.8) with observations across the model domain, but consistently exhibited a mean bias error (MBE) of 10+ ppbv on June 29th and July 1st (Figures S7 and S8). This high bias in O$_3$ production has been reported in other recent works (Travis et al., 2016; Schwantes et al., 2020; Place et al., 2023) which may be due to overestimated NO$_x$ emissions and/or improper representation of gas-phase organic chemistry.

Note that the 2017 EPA NEI used in this study is appropriate for a typical summer day and will likely not represent the actual emissions of the heatwave period caused by the stagnation event. Heatwaves can increase demand on the grid (Maia-Silva et al., 2020; Stone et al., 2023) and therefore increase $NO_x$ emissions due to greater combustion of fossil fuels from power generation (Chen et al., 2015), which are not represented by the 2017 NEI. Given the potential low bias in modeled $NO_x$ emissions, the high bias in modeled $O_3$ is even more perplexing, highlighting the complex chemistry involved in $O_3$ production. Importantly, the modeled MBE for $O_3$ is < 10 ppbv for central Missouri on June 29th, and Western New York on July 1st, locations that were upwind of WFM according to the HYSPLIT trajectories. This indicates that $O_3$ chemistry was well represented in the airmass that traveled to WFM.

PM2.5 model predictions performed worse compared to $O_3$ with many linear correlation values exhibiting null or negative values and MBE exceeding 10 µg m$^{-3}$. Similar to $O_3$, model MBE was < 10 µg m$^{-3}$ for Missouri and much of Chicago on June 29th and Western New York on July 1st. Three air quality monitoring sites in New York, measuring $O_3$, PM2.5, and 2 meter temperature, were chosen for time-series evaluations of WRF-Chem, including Pinnacle State Park (PSP) in the Southern Tier of New York, Queens College in New York City, and measurements at the old ski lodge below the summit of WFM (Figure S9). More information about the data collected at these sites can be found in Brandt et al. (2016) and Ninneman et al. (2020), while Pearson correlation values and MBE statistics can be found in Figure S9. WFM tends to show the lowest linear correlation with observations. This is likely due to WRF-Chem underestimating the elevation of WFM (1483m) by over 700m, and therefore not properly accounting for the topography in the region (Figure S10). PSP shows the lowest MBE values with high correlation coefficients (r > 0.7) for $O_3$ and 2m temperature. Finally, Queens college saw the strongest correlation coefficients for $O_3$ and 2m temperature (r > 0.85), but exhibited large positive biases for $O_3$ and PM2.5. The causes behind these overpredictions remain unclear but are beyond the scope of this work."

**Detailed comments:**

*Line 51: I think these are all modeling studies with the exception of Blando and Turpin, 2000 which is a literature review of plausibility. It would be good to cite as evidence a glyoxal–OxAc-cloud SOA reference that is experimental (lab or field).*

Observational/experimental studies were added to the list of citations including Sorooshian et al 2006, Carlton et al 2007, and Tan et al 2010.

*Line 94: "a manuscript regarding organic acid measurements is forthcoming" I find it difficult to assess some of the quantitative description in the absence of constraints on measurement uncertainty.*

Additional information was added to the methods sections that includes detection limits, which reads:

"While the exact detection limits of the organic acid analysis is currently being determined, a conservative estimate of 50 ug $L^{-1}$ for all 3 organic acids is used, based on the lowest concentration calibration standard. It is worth noting that the concentrations of the 3 organic acids investigated in this study are well above this conservative detection limit, with concentrations of 113, 111, and 23 times greater than the lowest concentration standard used in the calibrations respectively."

**Line 196: are these gas-phase or aqueous-phase photolysis rates? How were the different?**

Line 196 was updated to clarify gas phase photolysis rates. The $O_3$ and $H_2O_2$ aqueous phase photolysis rates are 1.5x greater than their gas-phase counterparts to represent the increased pathlength in the cloud droplets.

**The authors focus on a high-pressure stagnation event and seek to make associations with NOx. Electricity sector emissions increase during meteorological events like one studied here. Such events are often referred to high electricity demand events and there is increased reliance on peak load units. This may help the authors make their connections to NOx.**

This is a very interesting point brought up by the reviewer. Heat waves are well known to increase $NO_x$ emissions from the power sector. Additionally, the 2017 National Emissions Inventory is designed to represent a typical summer day and thus does not properly represent emissions from the power sector due to elevated temperatures. However, WRF-Chem overestimates $O_3$ mixing ratios in most regions of the domain during the pollution event, revealing a complex chemical system of potentially underestimated $NO_x$ emissions from the power sector occurring at the same time as high bias of modeled $O_3$ mixing ratios. We have added discussion of this phenomena to the WRF-Chem evaluation in section 3.2.2. The section reads:

"Modeled $O_3$ exhibited a strong positive linear correlation (r > 0.8) with observations, but consistently exhibited a mean bias error (MBE) of 10+ ppbv on June 29th and July 1st (Figures S7 and S8). This high bias has been reported in other recent works (Travis et al., 2016; Schwantes et al., 2020; Place et al., 2023) which may be due to overestimated $NO_x$ emissions and/or improper representation of gas-phase organic chemistry. Note that the 2017 EPA NEI used in this study is appropriate for a typical summer day and will likely not represent the actual emissions of the heatwave period caused by the stagnation event. Heatwaves can increase electrical demand (Maia-Silva et al., 2020; Stone et al., 2023) and therefore increase $NO_x$ emissions due to greater combustion of fossil fuels from the power generation (Chen et al., 2015), which are not represented by the 2017 NEI. At the same time, there is a high bias of modeled $O_3$ mixing ratios, highlighting the complex chemistry involved in $O_3$ production."

**Incorporating information used to Figure 8 into Figure 10 that directly compared concentrations from the observation-based estimates to the model predictions would be helpful.**

We are a bit unsure what the author is referring to with this comment. The blue and red dotted lines represent the observations made at Whiteface Mountain, allowing for direct comparison with model results. Additionally, information plotted in Figure 8 was used to initialize the gas+aqueous model that produced Figure 10. The first two sentences of Section 4.3 were edited to help clarify this:

"Cloud chemistry can alter organic acid concentrations distinct from gas phase chemistry alone. This section examines the impacts of aqueous chemistry by investigating both total mixing ratios and aqueous concentrations of HCOOH and $CH_3COOH$ using mixing ratios near WFM to initialize the model (Figure 10)."

**Line 390: I think it is more precise to start this sentence with "Model predictions suggest strong isoprene emissions …"**

Line 390 was updated to be more precise as the Reviewer suggested.

**Editorial:**

**Line 155: "PM 2.5" has a space and the number is not subscript**

This line was updated to include a subscript.

**Starting at line 172: "The complex geography of the Adirondack Mountains are likely not properly captured with a 12kmx12 km horizontal grid resolution …" "geography" is singular and "are" is plural. Also, I am sure there must be a paper describing the complex micrometeorology of WFM. Why not describe is more precisely?**

The word "are" was changed to "is". Unfortunately, there is very little research that has looked at the micrometeorology at WFM, but in general mesoscale flow includes mountain-valley winds and the mountain summits located above the PBL during night and morning and within the PBL during afternoons. In addition, when using a 12 km x 12 km horizontal grid mesh in WRF-Chem, the topography is not well represented. WRF-Chem underestimates the elevation of WFM by ~700m. Therefore, the mesoscale motions in WRF-Chem may not include upslope and downslope winds between 700 and 1500 m affecting the ability to compare properly between the model and observations at the summit. The authors have added a new figure to the supplemental material (Figure S10). Additionally, a sentence was added to the main text that reads:

"This is likely due to WRF-Chem underestimating the elevation of WFM (1483m) by over 700m, and therefore not properly accounting for the topography in the region (Figure S10). By using a 12 km x 12 km horizontal grid mesh in WRF-Chem, the topography is not well represented resulting in the modeled WFM summit to be underestimated by ~700 m and affecting the capabilities of WRF-Chem to represent mountain-valley winds and timing of when the summit is above and within the PBL (Giovannini et al 2020)."

***Many of the References are formatted incorrectly: Herckes et al., seems to have a personal note. Sometimes Atmos. Phys. Chem. references have "publisher: Copernicus GmbH", sometimes they do not.***

The citations in the reference section have been corrected. This seemed to be caused by strange errors within the first author's citation manager.

---

## Author Comment (AC2)

**Response to Reviewer 2**

We would like to thank the reviewer for their detailed edits and comments that have improved this manuscript. Below you will find our response to each comment/question. The reviewer's comments/questions are in bold italic font.

***General Comments The manuscript lacks a clear explanation regarding the impacts of its results and how it advances knowledge. In the abstract and introduction, there is discussion of the uncertainties associated with organic acids including how organic acids are commonly not included in detail in models and that few studies on organic acids in the Northeast U.S. exist. The manuscript then states "To address these shortcomings, this study investigates…". How are these shortcomings addressed with the results of this study? This is not clearly explained or highlighted since the conclusions seem to just restate the motivation of the study – there are uncertainties in organic acid production and further modeling and observational studies are needed. While future work can still be needed, it is important to address the significance of the results of this study that was performed and in what particular aspects it furthers current knowledge.***

We appreciate the comment, which prompted us to revise the abstract, introduction and conclusions to clarify significance of our study. We hope that we have made it clear that our study highlights the key processes affecting organic acid formation.

The changes are the following:

The abstract was updated to read:

[revised manuscript text omitted]

***The manuscript tends to use qualitative descriptive language and would benefit from more quantitative evaluations. For example, throughout the manuscript there are statements of "good model agreement", "significant reductions", "stronger production", "very little change", "substantially underestimated", "little correlation". How are these terms defined? What quantitative values do they represent? A statement such as "reduced by 80%" or "reduced with statistical significance by t-test" is more informative and stronger than "significant reductions". More quantitative results may also help in addressing the above concern.***

The authors agree that we were not quantitative enough while reporting results within the manuscript, particularly within the WRF-Chem evaluations section (Section 3). To address this

comment, we updated the WRF-Chem evaluation section to include new supplemental figures that show linear correlation coefficients and mean bias error (MBE) statistics for the WRF-Chem maps and these same statistics were applied to the WRF-Chem time series plots (now Figures S9) in the supplemental material. Section 3 was updated to read:

"Modeled $O_3$ exhibited a strong positive linear correlation ($r > 0.8$) with observations across the model domain, but consistently exhibited a mean bias error (MBE) of 10+ ppbv on June 29th and July 1st (Figures S7 and S8). This high bias in $O_3$ production has been reported in other recent works (Travis et al., 2016; Schwantes et al., 2020; Place et al., 2023) which may be due to overestimated $NO_x$ emissions and/or improper representation of gas-phase organic chemistry. Note that the 2017 EPA NEI used in this study is appropriate for a typical summer day and will likely not represent the actual emissions of the heatwave period caused by the stagnation event. Heatwaves can increase demand on the grid (Maia-Silva et al., 2020; Stone et al., 2023) and therefore increase $NO_x$ emissions due to greater combustion of fossil fuels from power generation (Chen et al., 2015), which are not represented by the 2017 NEI. Given the potential low bias in modeled $NO_x$ emissions, the high bias in modeled $O_3$ is even more perplexing, highlighting the complex chemistry involved in $O_3$ production. Importantly, the modeled MBE for $O_3$ is $< 10$ ppbv for central Missouri on June 29th, and Western New York on July 1st, locations that were upwind of WFM according to the HYSPLIT trajectories. This indicates that $O_3$ chemistry was well represented in the airmass that traveled to WFM.

Model predictions of PM 2.5 performed worse compared to $O_3$ with many linear correlation values exhibiting null or negative values and MBE exceeding 10 $\mu g \ m^{-3}$. Similar to $O_3$, model MBE was $< 10$ $\mu g \ m^{-3}$ for Missouri and much of Chicago on June 29th and Western New York on July 1st. Three air quality monitoring sites in New York, measuring $O_3$, PM2.5, and 2 meter temperature, were chosen for time-series evaluations of WRF-Chem, including Pinnacle State Park (PSP) in the Southern Tier of New York, Queens College in New York City, and measurements at the old ski lodge below the summit of WFM (Figure S9). More information about the data collected at these sites can be found in Brandt et al. (2016) and Ninneman et al. (2020), while Pearson correlation values and MBE statistics can be found in Figure S9. WFM tends to show the lowest linear correlation with observations. This is likely due to WRF-Chem underestimating the elevation of WFM (1483m) by over 700m, and therefore not properly accounting for the topography in the region (Figure S10). PSP shows the lowest MBE values with high correlation coefficients ($r > 0.7$) for $O_3$ and 2m temperature. Finally, Queens college saw the strongest correlation coefficients for $O_3$ and 2m temperature ($r > 0.85$), but exhibited large positive biases for $O_3$ and PM2.5. The causes behind these overpredictions remain unclear but are beyond the scope of this work."

We have also updated many sentences throughout the manuscript to be more quantitative when discussing model results.

**Specific/Technical**

**Comments Line 7: remove "analysi".**

This typo has been removed.

***Line 89: "form" should be "from".***

This typo has been removed.

***Line 251: Remove space before period.***

The space was removed.

***Line 251: Add space after "trajectories".***

A space was added after the word trajectories.

***Line 258: Add period at end of sentence.***

A period was added to the end of the sentence.

***Line 331-332: Remove "(left)" and "(right)". This is appropriate for the figure caption and seems unnecessary in the main body text.***

The words left and right were removed from the sentence.

***Line 340: What do these hour numbers represent? Please clarify.***

Figures 10 and 12 have been updated the x-axis to read "Local Time (EST)" for more clarity.

***Figure 1: Please make font size larger. Even with zooming in, some words are too small to be legible.***

An error in latex was causing the figure to appear smaller than the authors had originally intended. Figure 1 was enlarged to allow for easier reading.

***Figure 5: The caption references a) and b) but the images are not labeled as such.***

The figure caption was updated to remove the "a)" and "b)" references.

***Figures 7 & 8: In the caption, "Plum" should be "Plume".***

The word "plum" was changed to "plume" in each figure caption.

***Figure 9: What does the blue solid line represent?***

The blue line represents a fitted trend line using a generalized additive model (GAM) to estimate the average change of organic acid mixing ratios during the pollution event at WFM. A description of this trend line was added to the figure caption.

 ***Supplement: Section S3 comes before S2? Line 152 references Section S3 for WRF Chem description but it is listed as S2 in the supplement. "Figures S8" under Section S6 should be Figure S7. In Section S7, "Figure S9" should be Figure S8.***

The incorrect Section and Figure numbers have been corrected in the main text and supplemental material.